# Current State of Antimicrobial Treatment of Lower Respiratory Tract Infections Due to Carbapenem-Resistant *Acinetobacter baumannii*

**Marco Merli [1,*]** , **Federico D'Amico [1,2]** , **Giovanna Travi [1]** and **Massimo Puoti [1,3]**

[1] Infectious Diseases Clinic, ASST Grande Ospedale Metropolitano Niguarda, Piazza Ospedale Maggiore 3, 20162 Milan, Italy
[2] Post-Graduate School of Clinical Pharmacology, University of Milano, Via Festa del Perdono 7, 20122 Milan, Italy
[3] School of Medicine, University of Milano-Bicocca, Piazza dell'Ateneo Nuovo 1, 20126 Milan, Italy
* Correspondence: marco.merli@ospedaleniguarda.it

**Abstract:** Carbapenem-resistant *Acinetobacter baumannii* (CRAB) is a worldwide non-fermenting Gram-negative bacillus responsible for potentially severe nosocomial infections, especially in critically ill patients. CRAB tends to colonize inert surfaces and epithelia, especially the respiratory tract of mechanically ventilated patients, and may then become responsible for lower respiratory tract infections, probably the more challenging infection due to the site and the multidrug-resistant phenotype which makes it difficult to establish an effective antimicrobial regimen. Despite its diffusion, data regarding the treatment of CRAB are mainly retrospective and usually heterogeneous. Current international consensus guidelines prefer the use of ampicillin/sulbactam, but the strength of recommendation and grade of evidence tend to be weak to moderate. Moreover, no specific recommendation is given for different sites of infections. The recently introduced cefiderocol still received a recommendation against its use due to the results of the first randomized clinical trial, though retrospective and observational experiences showed favourable outcomes in this setting. We reviewed the major antibacterial drugs active against CRAB and discussed their combination in lower respiratory tract infections.

**Keywords:** *Acinetobacter baumannii*; antimicrobial therapy; pneumonia

## 1. Introduction

*Acinetobacter calcoaceticus-baumannii* complex (Acb) is a Gram-negative non-fermenting bacillus which has been associated with healthcare-associated infections worldwide, especially in critically ill patients. Its ability to acquire resistance to antimicrobials represents one of the major issues in its management [1].

The 2021 EARS-Net surveillance reported a high geographic variability in resistance among *Acinetobacter* species. The Baltic countries and Southern and South-Eastern Europe showed the highest prevalence of carbapenem resistance. In addition, a growing number of cases of *Acinetobacter* spp. infections have been reported in Europe recently, with a 43% increased prevalence from 2020 to 2021 [2].

A global survey conducted between 2004 and 2014 showed that *A. baumannii* is the microorganism most frequently exhibiting a multi-drug resistance (MDR) pattern, up to 64% in 2014 [3]. Similarly, the latest European Centre for Diseases Prevention and Control (ECDC) annual epidemiological report on antimicrobial resistance describes a combined resistance to fluoroquinolones, aminoglycosides and carbapenems in 36.8% of cases across Europe in 2021, far above the combined resistance rate of *Escherichia coli*, *Klebsiella pneumoniae*, *Pseudomonas aeruginosa*, *Staphylococcus aureus* and *Enterococcus faecium* [2]. The resistance pattern prompted the World Health Organization WHO in 2017 to list carbapenem-resistant *A. baumannii* (CRAB) among the major pathogens with critical priority for research and development [4]. A recent observational study found no difference in terms of outcome between CRAB and colistin-only-susceptible *A. baumannii* respiratory infections [5].

Five pathogenic species of the Acb (*Acinetobacter baumannii*, *Acinetobacter nosocomialis*, *Acinetobacter pittii*, *Acinetobacter seifertii* and *Acinetobacter dijkshoorniae*) have been identified, but—given the limited possibility of many laboratories to precisely distinguish species of the *calcoaceticus-baumannii* complex—for the purposes of this review we will use the designation *A. baumannii*, unless otherwise stated, in the broad sense to encompass all pathogenic members of the *Acb* complex. To note, *A. baumannii sensu strictu* was found associated with higher severity and mortality in nosocomial pneumonia with secondary bloodstream infection [6].

*A. baumannii* was initially considered a commensal opportunist, a low-virulence pathogen of little significance. The growing diffusion of invasive mechanical ventilation, the ubiquitarian use of central venous and urinary catheters, and the extensive administration of antibacterial therapy in critically ill patients favoured the surge of *A. baumannii* infections in terms of both severity and frequency [7].

Overall mortality rates in CRAB infections remain particularly high in all clinical studies, approaching 70% [8]. The most common clinical manifestations of CRAB are catheter-related bloodstream infections and nosocomial pneumonia, though surgical-site infections, post-surgical meningitis and ventriculitis and urinary tract infections have also been described. Differently from other opportunistic pathogens, which take advantage of the host's immune defects to develop and sustain the infection, the "opportunities" that *Acinetobacter* exploits are the disruption of anatomical barriers (i.e., severe burns, traumatic injuries, vascular and urinary catheters, endotracheal tubes) and the perturbation of normal host flora by exposure to broad-spectrum antibiotics which favour colonization [9,10].

Consequently, given the ability of CRAB to colonize surfaces and to survive by resisting disinfection and desiccation, the intensive care unit (ICU) became the hospital setting most deeply affected by *A. baumannii*. Nonetheless, the application of rigorous infection control measures has demonstrated the ability to eradicate CRAB also in ICU [11].

Nosocomial pneumonia, especially ventilation-associated pneumonia (VAP), is probably the most challenging clinical syndrome due to CRAB given its severity and the limited therapeutic options. The ability of *A. baumannii* to form biofilm communities on abiotic surfaces, such as endotracheal tubes, represents the first pathogenic step, fortunately not always followed by the dissemination to alveoli—favoured by mechanical ventilation—and the subsequent development of pneumonia [1].

The first step in the assessment of suspect *A. baumannii* pneumonia is usually represented by the need to discriminate respiratory tract colonization from infection. CRAB may be recovered from the respiratory specimen during surveillance cultures in a patient without current worsening of respiratory function nor evidence of lung infiltrates at imaging. Nonetheless, a respiratory deterioration would prompt a clinician to consider CRAB as a potential pathogen pending new cultures results.

The extreme flexibility of *Acinetobacter* in acquiring and expressing resistance mechanisms is the main responsibility of its extensively drug-resistant phenotype. Surface porins are frequently expressed. Carbapenem-resistance—the hallmark of extensive drug resistance—is usually mediated by the production of oxacillinases, such as OXA-24/40-like and OXA-23-like, but Metallo-β-lactamases and additional serine carbapenemases have also been recovered [12]. In particular, blaNDM-type genes were found to be located on either plasmid or chromosome in *A. baumannii* and the identification of a composite transposon Tn125 in both *A. baumannii* and *Enterobacterales* suggested a role of *Acinetobacter* in NDM transmission and diffusion to other species [13].

The antimicrobial strategy of CRAB pneumonia still lacks a standard of care. The choice of the molecules depends on both phenotypic and genotypic antimicrobial susceptibility tests, site of infection, patient's overall clinical status and organ functions, and finally possible concomitant bacterial infections. Interestingly, the consultation with an infectious disease specialist was not found to be associated with reductions in 30-day and 1-year all-cause mortality for CRAB infections, differently to what observed for other pathogens [14]. Nonetheless, the study included only a small number of patients with CRAB infections.

Table 1 summarizes the main clinical studies discussed in the treatment of CRAB.

**Table 1.** Summary of clinical studies.

| Author (Year) | NCT | References | DOI | Design | Treatment Phase | Group 1 | Group 2 | Primary Outcome | Effect | Effect | Risk of Bias |
|---|---|---|---|---|---|---|---|---|---|---|---|
| Betrosian Alex P. (2007) | NA | [15] | 10.1080/00365540600951184 | Open-label Prospective | NA | low-dose ampicillin-sulbactam (n = 14) | high-dose ampicillin-sulbactam (n = 13) | clinical improvement | 64.3% vs. 69.2%, (p = 0.785) | 64.3% vs. 69.2%, (p = 0.785) | M |
| Betrosian Alex P. (2008) | NA | [16] | 10.1016/j.jinf.2008.04.002 | Open-label Prospective | NA | ampicillin-sulbactam (n = 13) | Colistin (n = 15) | clinical improvement | 61.5% vs. 60% (NS) | 61.5% vs. 60% NS | M |
| Oliveira A.S. (2008) | NA | [17] | 10.1093/jac/dkn128 | Retrospective | NA | polymyxins (n = 82) | ampicillin/sulbactam (n = 85) | mortality | OR 2.07 (p = 0.041) | OR 2.07 (p = 0.041) | H |
| Altarac (2022) | NCT03894046 | [18] | 10.1093/ofid/ofac492.023 | Double-blind Randomized | III | Sulbactam-durlobactam (n = 63) | Colistin (n = 62) | all-cause mortality (28-day) | 19% vs. 32.3% | 19% vs. 32.3% | L |
| Demosthenes Makris (2018) | NA | [19] | 10.4103/ijccm.IJCCM_302_17 | Open-label Prospective | NA | colistin (n = 19) | colistin + ampicillin/sulbactam (n = 20) | clinical cure | 15.8% vs. 70%, (p = 0.001) | 15.8% vs. 70%, (p = 0.001) | M |
| Montero R. (2003) | NA | [20] | 10.1086/374337 | Open-label Prospective | NA | Colistin (n = 21) | Imipenem + cilastatin (n = 14) | clinical cure | 57% vs. 57% (NS) | 57% vs. 57% NS | M |
| Abdellatif S (2016) | NCT02683603 | [21] | 10.1186/s13613-016-0127-7 | Randomised, single-blind | IV | inhalatory colistin (n = 73) | intravenous colistin (n = 76) | clinical cure (VAP) | 67.1% vs. 72.3%, (p = 0.59) | 67.1% vs. 72.3%, (p = 0.59) | L |
| Paul M (2018) | NCT01732250 | [22] | 10.1016/S1473-3099(18)30099-9 | Open-label Randomized | IV | Colistin (n = 198) | Colistin + meropenem (n = 208) | Clinical failure | 83% vs. 81% (NS) | 83% vs. 81%, NS | L |
| Deng J (2022) | NA | [23] | 10.1186/s12879-022-07778-5 | Retrospective | NA | Tigecycline (n = 118) | Tigecycline + Sulbactam (n = 100) | Mortality (28-day) | 54.5% vs. 18.1%, (p < 0.001) | 54.5% vs. 18.1%, (p < 0.001) | H |
| Amat T (2018) | NA | [24] | 10.1016/j.cmi.2017.09.016 | Retrospective | NA | Colistin (n = 76) | colistin + tigecycline (n = 42) | crude mortality (30-day) | 62% vs. 57%, (p = 0.696) | 62% vs. 57%, (p = 0.696) | M |
| Ye J (2016) | NA | [25] | 10.1186/s12879-016-1717-6 | Retrospective | NA | Tigecycline (n = 84) | Sulbactam (n = 84) | Mortality (30-day) | 66.7% vs. 66.7% (NS) | 66.7% vs. 66.7% NS | M |
| Wunderink R (2021) | NCT03032380 | [26] | 10.1016/S1473-3099(20)30731-3 | Double-blind Randomized | III | cefiderocol (n = 148) | meropenem (n = 152) | all-cause mortality (14-day) | 0% vs. 46%, (p = 0.002) | 0% vs. 46%, (p = 0.002) | L |
| Falcone M (2022) | NA | [27] | 10.1128/AAC.00065-22 | Retrospective | NA | cefiderocol (n = 47) | Colistin (n = 77) | Mortality (30-day) | 55.8% vs. 34%, (p = 0.018) | 55.8% vs. 34%, (p = 0.018) | M |

**Table 1.** *Cont.*

| Author (Year) | NCT | References | DOI | Design | Treatment Phase | Group 1 | Group 2 | Primary Outcome | Effect | Effect | Risk of Bias |
|---|---|---|---|---|---|---|---|---|---|---|---|
| Bassetti M (2021) | NCT02714595 | [28] | 10.1016/S1473-3099(20)30796-9 | Open-label Randomized | III | cefiderocol (*n* = 16) | best available therapy (*n* = 37) | clinical cure | 43% vs. 27% | 43% vs. 27% | M |
| Russo A (2021) | NA | [29] | 10.6084/ m9.figshare.13056014 | Prospective | NA | Regimen with Fosfomycin (*n* = 44) | Regimen without Fosfomycin (*n* = 136) | Mortality (30-day) | 15.9% vs. 69.1%, (*p* < 0.001) | 15.9% vs. 69.1%, (*p* < 0.001) | H |
| Park J (2021) | NA | [30] | 10.3390/antibiotics10080903 | Retrospective | NA | Meropenem + Colistin (*n* = 66) | Meropenem + Tigecycline (*n* = 24) | Mortality (28-day) | 40.9% vs. 20.8%, (*p* = 0.078) | 40.9% vs. 20.8%, (*p* = 0.078) | M |

NA = not available; M = moderate; H = high; L = low.

## 2. Antimicrobial Agents

Among the antibacterial drugs which have been considered in a previous study for CRAB treatment, we will consider sulbactam—alone or in combination with the new beta-lactamase inhibitor durlobactam—, polymyxins, tigecycline, cefiderocol, and fosfomycin, Their site of action is described in Figure 1.

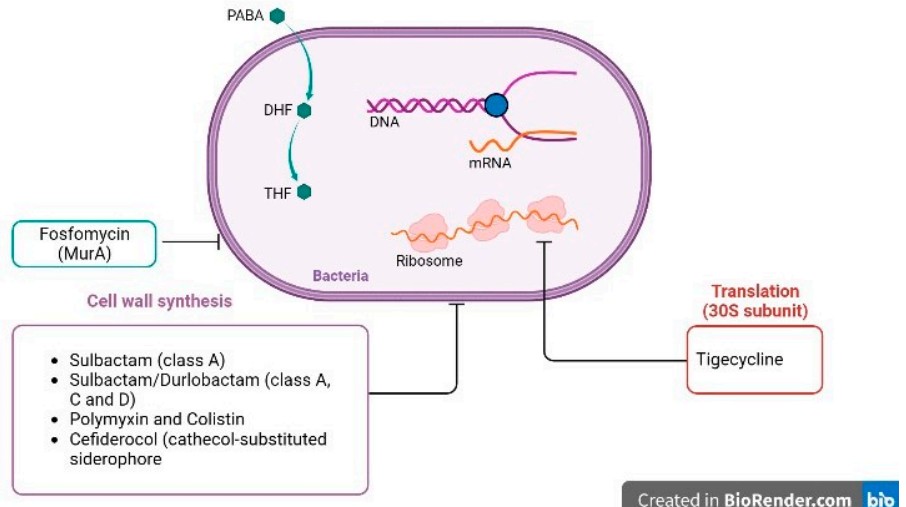

**Figure 1.** Antimicrobial agents for carbapenem-resistant *A. baumannii* and their mechanism of action.

## 3. Sulbactam

Sulbactam is a semi-synthetic penicillanic acid with intrinsic activity against *Acinetobacter*. As a first-generation beta-lactamase inhibitor, it is active only on a subset of Ambler class A beta-lactamases, such as PBP1a, PBP1b, and PBP3, required for the synthesis of bacterial peptidoglycan [31].

Resistance can be mediated by point mutations in PBP3 protein [32] and by degradation by acquired and upregulated beta-lactamases, such as TEM-1, ADC-30 and OXAs [33,34]. Moreover, metallo-beta-lactamases inactivate sulbactam [31], though they are currently rare in *A. baumannii* [35].

Previous pharmacokinetic studies showed an acceptable diffusion of sulbactam in alveolar lining fluid compared to plasma with a ratio of 0.61, which increased to 0.69 in patients with pneumonia [36].

In vitro activity against *Acinetobacter* spp. has been documented [37,38], though in vivo data are less conclusive. Moreover, EUCAST does not provide sulbactam breakpoints due to insufficient evidence [39], while CLSI defined susceptibility to ampicillin/sulbactam up to 8/4 mg/L and resistance from 32/16 mg/L [40].

An early randomized controlled trial explored the use of high-dose ampicillin/sulbactam (27 g vs. 36 g daily dose) in VAP sustained by MDR *A. baumannii*, reporting a 66.7% clinical improvement and 77.8% microbiological cure [15]. In a randomized controlled trial, ampicillin/sulbactam was found comparable to colistin in patients with VAP in terms of efficacy but exhibited a lower rate of nephrotoxicity 15% vs. 33%, as expected [16]. On the contrary, a retrospective analysis of invasive infections sustained by CRAB—including pneumonia—observed that treatment with polymyxin was associated with higher mortality compared to ampicillin/sulbactam (OR 2.07) [17]. In another retrospective analysis comparing intravenous colistin with ampicillin/sulbactam in CRAB-sustained VAP, no difference was observed in a 7-day clinical cure (47% vs. 56%, *p*=0.34), while a significantly higher 7-day microbiological failure (48% vs. 18%) and 30-day mortality (adjusted odds ratio 6.5) were documented in the colistin group [41]. Interestingly, the addition of ampicillin/sulbactam to colistin increased the early cure rate from 15.8% to 70% (*p* = 0.001) in a randomized controlled trial comparing colistin to colistin plus ampicillin/sulbactam in CRAB-sustained VAP [19]. In a small case series of pan-drug resistant *A. baumannii* VAP

treated with high-dose ampicillin/sulbactam, colistin (both intravenous and inhalatory) and high-dose tigecycline a favourable outcome was observed in 90% of patients [42]. Finally, in a recent meta-analysis high dose sulbactam (above 6 g per day) combination therapy had the highest ranking in clinical improvement and clinical cure [43], and a previous network meta-analysis also showed a better performance in mortality and clinical cure of sulbactam monotherapy compared to colistin (both monotherapy and combination therapy) and tigecycline [44].

Considering the present data, high-dose ampicillin/sulbactam has been recommended as first-line treatment by ESCMID and IDSA in VAP sustained by CRAB [45,46], though combination therapy is recommended when in vitro susceptibility is not demonstrated [45].

### 4. Sulbactam/Durlobactam

The high-susceptibility of sulbactam to hydrolysis by beta-lactamase prompted the development of the new generation diazabicyclooctane durlobactam, a potent inhibitor of Ambler class A, C and especially D enzymes, which are prevalent in *Acinetobacter* spp. Nonetheless, it has no activity on metallo-beta-lactamases [47].

The combination of sulbactam/durlobactam demonstrated in vitro efficacy against *A. baumannii*. Global isolates tested between 2016 and 2017 exhibited a $MIC_{50}/MIC_{90}$ $MIC_{90}$ (MIC, minimal inhibitory concentration) of 1/2 mg/L for sulbactam/durlobactam compared to $MIC_{50}/MIC_{90}$ of 8/64 mg/L for sulbactam alone, without significant geographic variation nor differences among subsets of resistance phenotypes and sources of infections. Moreover, sulbactam/durlobactam appeared superior to all tested comparators in vitro, showing similar potency to colistin [48].

The ATTACK is a Phase 3, multinational, randomised, controlled, noninferiority trial conducted to evaluate the efficacy and safety of sulbactam/durlobactam versus colistin, both in combination with imipenem/cilastatin as background therapy, for patients with serious infections due to *A. baumannii*, including CRAB strains. Patients with bloodstream and respiratory infections were included. All-cause mortality was significantly lower in the sulbactam/durlobactam group compared to the colistin group (19% vs. 32.3%) and higher clinical cure (61.9% vs. 40.3%) and more favourable microbiological outcome (68.3% vs. 41.9%) were observed. The better outcome of sulbactam/durlobactam was also confirmed among patients with CRAB [18].

### 5. Polymyxin

Polymyxins are a family of antibacterial agents which includes polymyxin B and colistin. They have been widely used in the treatment of CRAB infections given the low prevalence of resistance, often resulting a single agent being found susceptible in extensive drug-resistant *A. baumannii*. Nonetheless, the low diffusion of colistin in epithelial lining fluid after intravenous administration limits its use in lower respiratory tract infections [49–51]. A more favourable pharmacokinetic profile has been described for polymyxin B [52,53], though most of the available literature on the treatment of CRAB refers to colistin. Unfortunately, the risk of nephrotoxicity—which has been frequently reported at therapeutic doses—and the subsequent very narrow therapeutic window have limited their use [54].

A previous observational study found no difference between imipenem and intravenous colistin in VAP sustained by CRAB [20]. The addition of meropenem to colistin did not add any benefit in a randomized controlled trial including severe CRAB infections, among which almost a half were VAP [22]. Two recent meta-analyses found better outcomes with colistin compared to tetracyclines [43,44]. Direct comparison of colistin with ampicillin/sulbactam favoured the latter in terms of mortality, clinical cure and tolerability, especially considering the significantly higher incidence of nephrotoxicity with polymyxins [17,41,55,56].

The use of inhaled colistin was developed to overcome the low diffusion in the epithelial lining fluid, in order to provide adequate delivery of molecules in the distal airways, especially using vibrating mesh nebulizers [52]. Inhalatory administration provides higher drug concentrations in the epithelial lining fluid, together with lower systemic exposure and subsequently reduced incidence of nephrotoxicity [52,57]. In a prospective randomized trial in VAP sustained by Gram-negative bacilli (more than 50% were *A. baumannii*), inhalatory colistin showed a shorter time to bacterial eradication, increased p/F ratio and accelerated weaning from the ventilator (mean difference of 5 days). Moreover, a lower incidence of acute kidney injury compared to intravenous colistin was reported (17.8 vs. 39.4%, $p = 0.004$) [21]. Similar results were confirmed in meta-analysis [58,59]. Furthermore, the addition of aerosolized to intravenous colistin improved both bacterial eradication and clinical cure compared to intravenous administration alone in Gram-negative pneumonia [60].

The heterogeneity of clinical studies in terms of infectious agents and way of administration makes it difficult to draw firm conclusions. Overall, polymyxins have been considered second-line agents in the treatment of CRAB-sustained VAP [45,46]. Recommendations on the use of inhalatory antibacterials are still controversial. The latest ESCMID position paper recommends against the use of inhalatory antibiotics due to the low evidence of efficacy and the potential for respiratory adverse events [61]. Similarly, the IDSA consensus on MDR pathogens recommends against the use of nebulized antibiotics in patients with *A. baumannii* VAP [46], while the previous guidelines on VAP considered their use though with low quality of evidence and weak recommendation [62]. Finally, the consensus on the optimal use of polymyxins is that their use is considered appropriate in patients with VAP, especially those sustained by extensively drug-resistant pathogens, placing a high value on pharmacological considerations [63].

## 6. Cefiderocol

Cefiderocol, a novel catechol-substituted siderophore cephalosporin, was recently approved and introduced in clinical practice for the treatment of serious carbapenem-resistant Gram-negative infections. A pharmacokinetic study showed adequate concentrations in the epithelial lining fluids for MIC up to 4 mg/L in mechanically ventilated patients [64].

Conflicting data have been reported on its efficacy in the treatment of *A. baumannii*-sustained infections. The phase 3 randomized clinical study CREDIBLE-CR compared cefiderocol to the best available therapy in the treatment of carbapenem-resistant Gram-negative infections, showing an unexpectedly higher mortality with cefiderocol (49% vs. 18%) in the subgroup of *A. baumannii*-sustained infections [28]. The results may partly be explained by a higher prevalence of septic shock, renal dysfunction and pulmonary disease and a higher SOFA in the cefiderocol group, together with a lower mortality in patients with CRAB treated with best-available therapy compared to previous reports (18% vs. 40–50%). In the APEKS-NP trial, which evaluated cefiderocol vs. meropenem in nosocomial pneumonia, all-cause mortality did not differ among treatment arms (19% vs. 22%). Nonetheless, when considering patients with *Acinetobacter* spp. and meropenem MICs > 64 mg/L all-cause mortality was higher with meropenem compared to cefiderocol (0% vs. 46% at day 14 and 20% vs. 64% at day 28) [26]. In a retrospective series comparing cefiderocol-based vs. colistin-based regimens for CRAB infections (VAP 25.5%, $n = 12$ patients treated with cefiderocol vs. $n = 27$ with colistin) a better outcome with cefiderocol was observed in bloodstream infections (14-day and 28-day mortality 7.4% and 25.9% vs. 42.3% and 56.7%, respectively) but not in VAP (14-day and 28-day mortality 33.3% and 58.3% vs. 52.2% and 56.6%, respectively) [27]. Similarly, another retrospective study on CRAB infections in intensive care units (41% lower respiratory tract infections) did not show differences in mortality between cefiderocol- and colistin-based regimens [65]. To note, suboptimal cefiderocol exposure has been associated with clinical and microbiological failure especially in VAP and VAP-associated bloodstream infections [66], suggesting the potential for improving clinical success rate by optimizing drug administration. Several issues on cefiderocol use in critically ill patients need further clarification, especially when

treating CRAB infections: the PK/PD characteristics in patients with renal impairment and continuous renal replacement therapy, the use as monotherapy or in combination with other drugs, and its penetration into the ELF in patients with VAP.

Even though epidemiological studies showed susceptibility to cefiderocol in over 90% of *A. baumannii* isolates in different countries [67,68], resistance has been described due to the expression of PER-1 beta-lactamase, mutations in penicillin-binding proteins and down-regulation of iron transporters in CRAB [25]. Moreover, heteroresistant subpopulations have been detected after cefiderocol exposure [69].

Waiting for further clinical data, the ESCMID made a conditional recommendation against cefiderocol use in the treatment of CRAB VAP due to insufficient evidence [46], while IDSA reserved its use in case of infections refractory to other antibiotics or of intolerance or toxicity precluding their use [45].

## 7. Tigecycline

Since its introduction in clinical practice, tigecycline has shown considerable activity—though not universal—on *A. baumannii* spp. [70].

Clinical studies on tigecycline for the treatment of CRAB are heterogeneous in terms of the type of infection and antimicrobial combination. The addition of tigecycline (administered at a standard dose, 100 mg per day) to colistin did not demonstrate to improve mortality compared to colistin alone in CRAB bloodstream infections in critically ill patients (64% with VAP) in an observational study [24]. On the contrary, improved survival was observed in patients treated with sulbactam and tigecycline for CRAB pneumonia [23]. A retrospective analysis of MDR *A. baumannii* infections (54.4% respiratory tract) reported similar mortality with tigecycline compared to other regimens (36.1% vs. 38.3%) but a higher rate of favourable clinical outcomes in tigecycline treated patients (69.2% vs. 50%, $p < 0.001$); moreover, in multivariable analysis, both tigecycline alone and in combination had a reduced risk of unfavourable outcomes (odd ratio 0.47 and 0.55, respectively) [71]. Similar efficacy of tigecycline-based to sulbactam-based regimens was reported in a retrospective cohort of critically ill patients with MDR *A. baumannii* infections (70% bilateral pneumonia) [72]. Other retrospective studies in critically ill patients with pneumonia reported a worse outcome with tigecycline (standard dose) compared to other regimens, but the excess mortality of tigecycline is significant only among patients with MIC > 2 mg/L [73], or similar efficacy colistin-based regimens [74]. A meta-analysis of available studies (no randomized-controlled trial was available, and most studies were conducted in China) evaluated tigecycline in the treatment of MDR *A. baumannii* pneumonia, confirming similar efficacy of tigecycline-based to other regimens, but a lower microbiological eradication rate and a lower risk of nephrotoxicity compared to colistin-based regimens [75].

Even though most published studies report the use of standard-dose tigecycline, a double dose was found to reach therapeutic concentrations in the ELF with potential clinical success in 94–100% to 41–75% of cases with MIC 0.12–0.25 to 0.5–1 mg/L [76,77] and could help to reduce microbiological failure and prevent the emergence of resistance and heteroresistance. Finally, high-dose tigecycline was also ranked better than colistin monotherapy for clinical outcomes in a network meta-analysis [44].

High-dose tigecycline and minocycline have been considered as second-line agents, as part of combination therapy, in severe CRAB infections [45,46].

## 8. Fosfomycin

Fosfomycin is a phosphoenolpyruvate analogue which acquired progressive relevance in the treatment of severe infections sustained by Gram-negative bacilli in the last decade.

Even though *A. baumannii* is intrinsically resistant to fosfomycin, an in vitro study observed that fosfomycin in combination with sulbactam (4 g every 8 h) displayed synergism in 74% of *A. baumannii* isolates, resulting in a median MIC50 and MIC90 reduction respectively of 4–8-fold compared to monotherapy [78]. A randomized controlled trial

explored the combination of colistin and fosfomycin compared to colistin monotherapy in severe CRAB infections (almost 75% were pneumonia in both arms), showing a higher microbiological eradication with combination therapy, but only a mild trend to better clinical outcome [79]. A retrospective analysis of CRAB pneumonia found an independent association between the use of combination fosfomycin therapy and 30-day mortality [29]. Further clinical data are limited to a case report of successful treatment of post-surgical meningitis with ampicillin/sulbactam, rifampin and fosfomycin [80] and a case series reporting a favourable outcome of the association of fosfomycin with cefiderocol in severe CRAB infections (*n* = 5, only one patient had a VAP) [81].

Currently, no recommendation has been made regarding its use in clinical practice, though—considering the interesting in vitro observation—its use in combination therapy could become a therapeutic option if supported by clinical data.

## 9. Combination Therapy

Combination therapy has been evaluated in both prospective and retrospective studies, even though the regimens are usually heterogeneous making difficult to draw firm conclusions on the significant clinical benefit of combination therapy compared to monotherapy. No benefit of combination was observed by associating meropenem with colistin in CRAB infections, given the usual high-level resistance to carbapenems. Moreover, despite evidence of in vitro synergism of rifampin with colistin, clinical data did not document a beneficial effect on survival of the combination regimen compared to colistin monotherapy in CRAB VAP, even though a trend to higher microbiological eradication has been observed [30,82,83].

A network meta-analysis supports the use of sulbactam-based combination therapy for ranking in clinical improvement and clinical cure [43], as described above.

Fosfomycin has been considered as a partner in combination therapy due to the in vitro observation of sulbactam MIC reduction [78]. A previous study in a small sample reported a potential—though not significant—benefit with fosfomycin plus colistin combination compared to colistin monotherapy in terms of microbiological success [79]. Fosfomycin has been included in combination therapies with cefiderocol in recent case series, but no superiority to other regimens has been demonstrated [27,81].

Despite the low quality of evidence, combination therapy is widely used to treat severe CRAB infections—as also endorsed by IDSA recommendations [45]—since it is often difficult to confirm in vitro susceptibility to drugs other than polymyxins in routine clinical practice and limited data support the efficacy of any molecule against CRAB. Nonetheless, combination therapy appeared to drive better outcomes when two active drugs were combined, such as colistin and sulbactam [46]. More data on combination regimens including cefiderocol and/or fosfomycin are yet to come.

## 10. Risk of Bias Assessment

The risk of bias in each study was assessed by two reviewers independently using the adapted versions of the Effective Practice and Organization of Care guidelines for RCT and the Newcastle Ottawa Scale for non-randomized studies [84,85]. Studies were overall classified as providing low, moderate, or high risk of bias evidence.

## 11. Discussion

The antimicrobial therapy of CRAB-sustained VAP remains a challenging issue. The limited pathogenicity of *Acinetobacter* together with the critical illness and the usual complex clinical condition of patients with CRAB infections make it difficult to discriminate infection from other concomitant morbid conditions as determinants of the outcome. Consequently, the results of both retrospective and prospective studies may be difficult to interpret. Moreover, many retrospective and observational studies pool data of heterogenous infectious syndromes, creating conflicting data on the impact of antibacterial agents and combination regimens for CRAB infections in different syndromes.

High-dose sulbactam appears to be the most effective treatment option for CRAB VAP. The benefit of combination with colistin, as also suggested by ESCMID recommendations [46], is counteracted by the potential for severe nephrotoxicity and the limited drug penetration in the epithelial lining fluid after intravenous administration. Nebulized colistin administration could thus be considered to prevent toxicity and improve drug delivery to epithelial lining fluid if an adequate nebulization system is available, even though data regarding its efficacy are discordant. The use of vibrating membrane devices, which allow the delivery of drugs in distal airways, especially in ventilated patients, could increase the effectiveness of colistin in this difficult-to-reach site.

Currently, available data do not support the use of cefiderocol in VAP sustained by CRAB, though it is one of the very limited treatment options in MBL-producing CRAB. Very few clinical data are yet available in this context. Nonetheless, since adequate cefiderocol exposure in the site of infection—especially the epithelial lining fluid—has been reported as a relevant step in achieving antibacterial efficacy in critically ill patients with renal replacement therapy [66], improvement in drug administration to achieve optimal exposure needs to be considered to correctly evaluate cefiderocol efficacy in CRAB VAP.

Given the limited availability in clinical practice of in vitro susceptibility of CRAB to most antimicrobials except colistin, combination therapy is currently widely used and appears advisable in critically ill patients with deep-seated infections. Depending on the site of infection, tigecycline and colistin may become—mostly alternative—partners based on their different pharmacokinetics properties. Especially high-dose tigecycline showed efficacy in difficult-to-treat pneumonia, while colistin could be considered as intravenous administration in VAP-associated bloodstream infection or as nebulized administration in lower respiratory tract infections if an adequate nebulization system is available.

The limited and heterogenous data available do not allow to draw more definite considerations. Nonetheless, the expected availability of sulbactam/durlobactam in the near future will add another pharmacological tool, potentially replacing sulbactam with expected higher activity [18], though activity will still be lacking in metallo beta-lactamase expressing CRAB.

## 12. Conclusions

CRAB has been reported as an opportunistic pathogen in critically ill patients with pneumonia, though its role and weight in determining the patient's outcome and mortality are yet to be defined. While the treatment of CRAB in patients with VAP with associated bloodstream infection is mandatory, it is difficult to discriminate the role of *A. baumannii* in patients with VAP without bloodstream infection.

Given the potential adverse events and further selection of antimicrobial resistance, the decision of treatment and the choice of the antimicrobial regimens should be customized for each patient based on its clinical condition, imaging, laboratory evidence and microbiological—both culture and molecular methods—results. As for all other infections, treatment needs to be re-evaluated daily for indication and duration.

If a respiratory infection is confirmed or highly suspected in critically ill patients, combination therapy is currently recommended until further evidence becomes available with the most recent molecules, such as cefiderocol and sulbactam/durlobactam.

## 13. Future Directions

The recent availability of cefiderocol and the expected availability of sulbactam/durlobactam brought expectations in the management of CRAB infections, especially ventilation-associated pneumonia. Nonetheless, given the pathophysiology of this infection and its "opportunistic" behaviour, it will be crucial to better define the patient's CRAB-related outcome and to accurately distinguish infection from colonization, aiming at more reliable results to guide clinical practice. Moreover, especially for beta-lactams, the optimization of drug dosing and administration in terms of PK/PD may help improve patients' outcomes and it is thus mandatory in severely ill patients, also to correctly evaluate the efficacy.

**Author Contributions:** Conceptualization, M.M. and F.D.; methodology, M.M., F.D. and M.P.; data curation, M.M., F.D. and G.T.; writing—original draft preparation, M.M. and F.D.; writing—review and editing, M.P. and G.T.; supervision, M.P. All authors have read and agreed to the published version of the manuscript.

**Funding:** No funding was received for this review.

**Institutional Review Board Statement:** Not applicable.

**Informed Consent Statement:** Not applicable.

**Data Availability Statement:** Not applicable.

**Conflicts of Interest:** The authors declare no conflict of interest.

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
