# Peer review of "Current State of Antimicrobial Treatment of Lower Respiratory Tract Infections Due to Carbapenem-Resistant Acinetobacter baumannii"

_futurepharmacol, doi:10.3390/futurepharmacol3020030_

Round 1

Reviewer 1 Report

This is an exceleent review on CRAB pneumonia management and available antibiotics. I only have one comment:

Lines 206 and on: In fact, the latest IDSA guidelines on antimicrobial resistance of 2022 recommended against the use of inhalational polymixins. The HAP/VAP guidelines were published in 2016. So, this new updated recommendation in the 2022 AMR guidelines is what should the authors refer to. Hence, the sentence should start with something like "Similarly" instead of "On the contrary" and then state the new recommendation against the use of inhalation antibiotics and state that this is an update to the old recommendation in the HAP/VAP guidelines that inhalation route could be used. See: https://www.idsociety.org/practice-guideline/amr-guidance-2.0

Author Response

Dear reviewer, thank you for your comment. The sentence was revised according to your consideration.

Reviewer 2 Report

1. The English language of the manuscript should be revised.

2. The abbreviations should be first mentioned in complete words when first mentioned in the manuscript.

3. The declarations section in the end of the manuscript should be written

4. The mechanism of action of the drugs should be presented in a figure.

5. More detailed conclusion should be written

6. The table should be modified and DOI should be replaced with the reference number and abbreviations should be clarified in a table caption.

The English language of the manuscript should be revised.

Author Response

Dear Reviewer,

thank you for your comments. Here is a point-by-point answer to your issues

1. The English language of the manuscript has been fully revised.

2. The abbreviations have been checked, extense form has been provided

3. The declarations section in the end of the manuscript has been completed

4. The mechanism of action of the drugs has been presented in a figure.

5. The conclusions have been reformulated

6. The table was modified replacing DOI with reference numbers. Abbreviations were clarified in a footnote.

Reviewer 3 Report

I have the following comments and questions for the authors. There are many awkward phrases that I do not point out here; I only point out those where the meaning cannot be interpreted:

Please double check the article by a native English reader.

First paragraph from Polymyxin  is not clear can be rewrite in more clear format.

The conclusion need to clear and specific.

Disccusion is to short extend!

My recommendation is to focus on short conclusion!

Please recheck the References order!

Please double check the article by a native English reader.

Author Response

Dear Reviewer,

thank you for your comments. Here you can find a point-by-point answer

  • English language has been fully revised
  • The first paragraph on polymyxins has been revised
  • Reference order has been double checked

Round 2

Reviewer 2 Report

The manuscript can be accepted in the current form

Reviewer 3 Report

Accept in current format!

English style is fine.